# Thermo-Responsive ZnPc-*g*-TiO_2_-*g*-PNIPAM Photocatalysts Sensitized with Phthalocyanines for Water Purification under Visible Light

**DOI:** 10.3390/molecules27103330

**Published:** 2022-05-22

**Authors:** Bingxin Mao, Cong Liu, Xu Cui, Yanhui Li, Qian Duan

**Affiliations:** 1School of Materials Science and Engineering, Changchun University of Science and Technology, Changchun 130022, China; bingxinmao123@163.com (B.M.); liucong123456@xmu.edu.cn (C.L.); lyh2008@cust.edu.cn (Y.L.); 2Engineering Research Center of Optoelectronic Functional Materials, Ministry of Education, Changchun 130022, China

**Keywords:** photocatalyst, thermo-responsive, titanium dioxide, phthalocyanine, *N*-isopropyl acrylamide

## Abstract

A novel thermo-responsive 2,9(10),16(17),23(24)-tetrakis[(3-carboxyacrylamide) phthalocyaninato] zinc (ZnPc)-*g*-TiO_2_-*g*-poly(*N*-isopropylacrylamide) (PNIPAM) photocatalyst modified with phthalocyanines was prepared. The photocatalyst exhibited thermo-responsive properties due to the introduction of PNIPAM, which performed recovery for reuse above the lower critical solution temperature (LCST, about 26 °C). ZnPc-*g*-TiO_2_-*g*-PNIPAM effectively expanded the light response range to the visible light region and inhibited the recombination of electron–hole pairs, which enhanced the performance of the photocatalyst. As expected, ZnPc-*g*-TiO_2_-*g*-PNIPAM (0.3 g/L) exhibited excellent photocatalytic performance for the removal of Rhodamine B (RhB, 1.0 × 10^−5^ mol/L) and methylene blue (MB, 1.0 × 10^−5^ mol/L) under visible light, which reached 97.2% and 88.6% at 20 °C within 40 min, respectively. Furthermore, the influence of temperature upon photocatalytic performance was also investigated. When the temperature increased from 20 °C to 45 °C, the removal of RhB decreased by approximately 53.8%. The stability of the photocatalyst demonstrated that the photocatalytic activity was still above 80% for the removal of RhB after 3 cycles. Above all, this work provided an intelligent thermally responsive photocatalyst based on phthalocyanine for water purification under visible light.

## 1. Introduction

In recent years, industrial development has led to undesirable environmental problems around the world, especially water pollution, causing serious health consequences for all living beings [1,2]. As a green and energy-saving technology, photocatalysis has been considered to be one of the most effective methods in environmental protection by means of the utilization of solar energy, especially in eliminating hazardous pollutants in water [3,4]. The realization of photocatalytic technology depends on the preparation of high-performance photocatalysts [5,6,7]. As the first reported photocatalyst in 1972 [8], TiO_2_ has received an increasing amount of attention, owing to its chemical stability, low cost, nontoxicity and high photocatalytic activity [9,10]. Unfortunately, in addition to the above advantages, the large band gaps (3.2 eV), the limitation of visible light absorption and the fast recombination rate of electron–hole pairs still restrict the application of TiO_2_ in the field of photocatalysis [11,12].

In order to overcome the shortcomings of TiO_2_, various methods have been developed by doping, defecting, modifying surface and sensitizing [10]. Among the various possible strategies, sensitizing appears to be a good option due to the simple operating procedure and mild reaction [13]. Phthalocyanines (Pcs), as a kind of organic semiconductor, have a strong response at around 600~800 nm and can broaden the light response range of TiO_2_ to the visible light area [14], which can also accelerate the transmission of photogenerated electrons [15]. Pcs-sensitized TiO_2_ shows a higher degradation rate than pure TiO_2_; however, there are still some drawbacks which limit its further development, such as easy aggregation, which reduces catalytic efficiency, and inefficient recovery, inducing secondary pollution in water treatment.

As a thermo-responsive polymer, poly(*N*-isopropylacrylamide) (PNIPAM) possesses a lower critical solution temperature (LCST) of about 32 °C [16,17], and PNIPAM-grafted materials also exhibit temperature-sensitive properties. Above LCST, PNIPAM-grafted materials are hydrophobic, and they will push material chains out of the water to facilitate recovery, which can effectively overcome the above shortage of photocatalysts. Duan et al. reported PNIPAM/TiO_2_ nanoparticles with thermal responsibility, and their photocatalytic activity was enhanced by combining them with polymers [18]. Therefore, constructing a thermal-responsive photocatalyst will enable composites to have a broader application field. Herein, we prepared a novel phthalocyanine-based thermo-responsive ZnPc-*g*-TiO_2_-*g*-PNIPAM photocatalyst, as shown in Figure 1. PNIPAM and ZnPc were grafted on TiO_2_ by radical polymerization and chemical bonds, respectively. PNIPAM grafting possesses excellent thermo-responsive properties and ZnPc-sensitizing enhances photocatalytic activity for the pollutant removal. In addition, recyclability and photocatalytic mechanism were investigated. The results show that the recovery efficiency of ZnPc-*g*-TiO_2_-*g*-PNIPAM is improved while maintaining excellent photocatalytic performance and cycling stability. Above all, this work was expected to fabricate an effective, stable and recyclable photocatalyst for organic pollutants removal.

## 2. Results and Discussion

Figure 1A shows the XRD patterns of TiO_2_, TiO_2_-*g*-MPS, TiO_2_-*g*-PNIPAM and ZnPc-*g*-TiO_2_-*g*-PNIPAM. TiO_2_ exhibited high crystallinity and all the diffraction peaks were attributed to anatase TiO_2_ (PDF # 21-1272). After MPS, PNIPAM and ZnPc were grafted on TiO_2_, the prepared TiO_2_-*g*-MPS, TiO_2_-*g*-PNIPAM and ZnPc-*g*-TiO_2_-*g*-PNIPAM show the same diffraction peaks belonging to anatase TiO_2_ and no new diffraction peaks were found, indicating surface modification has little effect on the structure of TiO_2_.

FT-IR spectra were performed to further explore the composition of prepared samples, as shown in Figure 1B. Compared with TiO_2_, the absorption peaks of TiO_2_-*g*-MPS at 1320 and 1299 cm^−1^ were attributed to -C-CO-O- skeletal vibration, and C-H bending vibration was found at 1469 cm^−1^, which indicates that TiO_2_ was successfully modified by MPS. For TiO_2_-*g*-PNIPAM, the strong characteristic peaks at 1645 and 1542 cm^−1^ were attributed to -C=O vibration and N-H bending vibration, indicating PNIPAM grafted on TiO_2_. For ZnPc-*g*-TiO_2_-*g*-PNIPAM, the characteristic peak at 1651 cm^−1^ was overlapped with the amide I band of PNIPAM, and 1395 cm^−1^ was attributed to the -COOTi- bond, which was formed by the reaction between -OH on TiO_2_ and -COOH of ZnPc [19,20]. The above analysis shows that ZnPc and PNIPAM were successfully grafted on TiO_2_.

In order to further explore the structure of ZnPc-*g*-TiO_2_-*g*-PNIPAM, XPS spectra of ZnPc-*g*-TiO_2_-*g*-PNIPAM were examined. It can be found from Figure 2A that there are four main elements (Ti, C, N and O) in ZnPc-*g*-TiO_2_-*g*-PNIPAM. Figure 2B presents the Ti 2p peaks of ZnPc-*g*-TiO_2_-*g*-PNIPAM, with two peaks at 463.9 eV and 458.2 eV, respectively. The difference between the two peaks was 5.7 eV, indicating Ti^4+^ existed in ZnPc-*g*-TiO_2_-*g*-PNIPAM [21]. Figure 2C shows the C 1s peaks at 288.2 eV, 287.0 eV, 285.6 eV and 284.5 eV due to N-C=N, N-C=O, C-C=O and C-C, respectively [14,22]. The O1s peaks at 531.8 eV, 529.5 eV and 532.8 eV are observed in Figure 2D, corresponding to -OH species, lattice oxygen in TiO_2_ and C=O in PNIPAM. The N 1s peak at 398.3 eV was assigned to C-N for the PNIPAM-grafted TiO_2_ in Figure 2E. Meanwhile, the N 1s peaks at 400.0 eV and 399.1 eV were attributed to pyridyl N and pyrrolic N. The above results reveal that ZnPc and PNIPAM were successfully combined with TiO_2_.

The SEM images of TiO_2_, TiO_2_-*g*-PNIPAM and ZnPc-*g*-TiO_2_-*g*-PNIPAM were shown in Figure 3. In Figure 3A, TiO_2_ nanoparticles displayed uniform spherical morphology with the diameter of about 20–50 nm. Compared with TiO_2_, there was no obvious change for TiO_2_-*g*-PNIPAM and ZnPc-*g*-TiO_2_-*g*-PNIPAM in Figure 3B,C. Moreover, the elemental mapping image confirmed Ti, C, O and Zn elements in ZnPc-*g*-TiO_2_-*g*-PNIPAM (Appendix A), further proving that ZnPcs were distributed on ZnPc-*g*-TiO_2_-*g*-PNIPAM.

The thermal stability of the prepared samples by TG further confirmed the organic layer coating on TiO_2_ in Figure 4. Compared with TiO_2_, TiO_2_-*g*-MPS shows a weight loss of about 7.6 wt% when heating to 900 °C, due to the desorption of adsorbed water and decomposition of MPS. For TiO_2_-*g*-PNIPAM, there was obvious weightlessness at 290–440 °C originating from the decomposition of PNIPAM. Meanwhile, by comparing TiO_2_-*g*-PNIPAM and ZnPc-*g*-TiO_2_-*g*-PNIPAM at 800–900 °C, the tiny difference might be due to the ZnPc decomposition in ZnPc-*g*-TiO_2_-*g*-PNIPAM.

The specific surface areas and porous structures of TiO_2_ and ZnPc-*g*-TiO_2_-*g*-PNIPAM were investigated, as shown in Appendix A. The specific surface areas of TiO_2_ and ZnPc-*g*-TiO_2_-*g*-PNIPAM were 64 m^2^/g and 36 m^2^/g in Appendix A and their pore size distributions were mainly around 3–15 nm in Appendix A. Compared with TiO_2_, ZnPc-*g*-TiO_2_-*g*-PNIPAM had lower specific surface area, which might be due to the more crowded particle internal space with PNIPAM and ZnPc grafting.

UV-vis spectra of ZnPc in dimethyl sulfoxide shows B band at 361 nm and Q band at 699 nm (Appendix A), revealing a strong absorption of ZnPc in visible light region. The optical properties of TiO_2_ and ZnPc-*g*-TiO_2_-*g*-PNIPAM were also studied by UV-vis diffuse reflectance spectroscopy in Figure 5. The absorption of pure TiO_2_ was mainly in the UV region, and almost no absorption under visible light is observed in Figure 5A. Compared with TiO_2_, ZnPc-*g*-TiO_2_-*g*-PNIPAM shows strong absorption in the visible light region, indicating that ZnPc-sensitized TiO_2_ could effectively broaden the light response range. Meanwhile, Figure 5B shows the estimated band gaps of TiO_2_ and ZnPc-*g*-TiO_2_-*g*-PNIPAM, which are about 3.19 eV and 2.88 eV, respectively. Although pure TiO_2_ had a large band gap, ZnPc-sensitized TiO_2_ could significantly reduce the band gap and benefit the absorption of visible light.

In this study, the LSCT was arbitrarily defined as the temperature corresponding to a 10% decrease in transmittance (see Appendix A). The LCST of ZnPc-*g*-TiO_2_-*g*-PNIPAM was 26 °C (Appendix A), which is lower than PNIPAM homopolymers. It can be attributed to hydrophobic ZnPc existing in the photocatalyst, which decreases its hydrophilicity, resulting in the low LCST. When the temperature reached 45 °C (>LCST), the PNIPAM chain grafted on the surface of TiO_2_ became hydrophobic and ZnPc-*g*-TiO_2_-*g*-PNIPAM quickly separated from the solution in 120 min, as seen in Appendix A. Meanwhile, for TiO_2_, there was no obvious change at 45 °C in 120 min. Therefore, the introduction of PNIPAM facilitates the recovery of photocatalyst.

### 2.1. Photocatalytic Activity

The photocatalytic activities of prepared photocatalysts were mainly evaluated by the removal of RhB and MB. First, the removal efficiencies of RhB by different amounts of ZnPc-*g*-TiO_2_-*g*-PNIPAM with H_2_O_2_ (1 mL) under visible light in 40 min at 20 °C were studied. As shown in Figure 6A, the photocatalytic activity improved steadily with photocatalyst concentration increased to 0.3 g/L. However, when the concentration of ZnPc-*g*-TiO_2_-*g*-PNIPAM reached 0.4 g/L, the removal efficiency decreased, which can be attributed to the aggregation of photocatalysts and the reduction in effective active sites.

Figure 6B shows the removal efficiencies of RhB (4.8 mg/L) by blank, H_2_O_2_, ZnPc-*g*-TiO_2_-*g*-PNIPAM and ZnPc-*g*-TiO_2_-*g*-PNIPAM/H_2_O_2_ under visible light at 20 °C. It can be found that RhB was difficult to removed without photocatalysts, and the removal of RhB by pure H_2_O_2_ (1 mL) was only 27% in 40 min. Meanwhile, for TiO_2_, which did not absorb visible light, the removal efficiency of RhB could reach 25.8% under visible light in 40 min. That was because RhB can be excited by visible light, and when the concentration of RhB was high, photogenerated electrons could be transferred to TiO_2_, resulting in a low efficiency in the photocatalytic activity of TiO_2_ in RhB solution under visible light [23,24]. It can also be found that the removal of RhB was 97.2% after ZnPc-*g*-TiO_2_-*g*-PNIPAM (0.3 g/L) combined with H_2_O_2_ in 40 min under visible light, which was higher than ZnPc-*g*-TiO_2_-*g*-PNIPAM. More radicals were generated to remove RhB under visible light because photocatalysts absorbed visible light and transferred energy to H_2_O_2_.

Figure 6C also shows the removal of MB (8.0 mg/L) with prepared samples. The removal efficiency of H_2_O_2_, ZnPc-*g*-TiO_2_-*g*-PNIPAM and ZnPc-*g*-TiO_2_-*g*-PNIPAM/H_2_O_2_ was 24.7%, 20.9% and 88.6%, respectively, in 40 min under visible light at 20 °C. ZnPc-*g*-TiO_2_-*g*-PNIPAM/H_2_O_2_ also shows better photocatalytic performance than others. The effects of temperature on the visible-light photocatalytic properties of the ZnPc-*g*-TiO_2_-*g*-PNIPAM were investigated. Figure 6D shows the photocatalytic efficiency of ZnPc-*g*-TiO_2_-*g*-PNIPAM/H_2_O_2_ for RhB removal at different temperatures, including lower that (20 °C), similar to (38 °C) and significantly higher than (45 °C) LCST. It can be found that ZnPc-*g*-TiO_2_-*g*-PNIPAM at 20 °C exhibited better photocatalytic performance than 38 °C and 45 °C, and the removal efficiency of RhB at 45 °C in 40 min was 43.4%. This indicates that temperature variation can significantly affect the activity of photocatalytic performance for RhB removal, which might be due to PNIPAM chains becoming hydrophobic above LCST and covering the surface of TiO_2_, thus resulting in the reduction in active sites in the photocatalytic reaction. ZnPc-*g*-TiO_2_-*g*-PNIPAM also exhibited high photocatalytic activity for the removal of TC•HCl (10 mg/L), and the removal efficiency could reach 31.9% in 75 min under visible light (Appendix A). To investigate the kinetic behavior of RhB and MB photodegradation, experimental data were fitted with zero-order, first-order and second-order models. The R^2^ values are shown in Appendix A; the higher correlation coefficient of zero-order kinetics illustrates good agreement between the data and the equation [2].

ZnPc-*g*-TiO_2_-*g*-PNIPAM after the photocatalytic reaction was collected at 45 °C and washed with ethanol/water solution for 90 min then dried at 60 °C under vacuum for the next experiment. As shown in Figure 7A, ZnPc-*g*-TiO_2_-*g*-PNIPAM still maintained high photocatalytic activity after three cycles, and the decreased efficiency was mainly caused by the loss in the recovery process. Moreover, after three photocatalytic tests, FT-IR spectra (Figure 7B), the XRD pattern (Figure 7C) and the SEM image of ZnPc-*g*-TiO_2_-*g*-PNIPAM displayed no obvious changes compared with the pristine photocatalyst. Therefore, the above results show that ZnPc-*g*-TiO_2_-*g*-PNIPAM had good stability.

### 2.2. Mechanism Discussion

Figure 8A shows the mechanism during the process of RhB removal by using trapping agents of benzoquinone (BQ) and isopropanol (IPA) to capture O_2_**·**^−^ and OH**·**. It can be found that the photocatalytic activity decreased after adding BQ and IPA, indicating that O_2_**·**^−^ and OH**·** were the main active substances. Photoluminescence (PL) spectra were usually used to evaluate the recombination ability of photogenerated electron–hole pairs. As shown in Figure 8B, the PL peak intensity of ZnPc-*g*-TiO_2_-*g*-PNIPAM under excitation of 315 nm was lower than TiO_2_, which confirmed that ZnPc-sensitized TiO_2_ could effectively promote photoelectron migration. Moreover, EIS Nyquist plots of TiO_2_ and ZnPc-*g*-TiO_2_-*g*-PNIPAM were shown in Figure 8C, and the smaller arc radius indicated the higher transfer efficiency of photogenerated electron–hole pairs [25]. ZnPc-*g*-TiO_2_-*g*-PNIPAM exhibited a smaller arc radius than TiO_2_, revealing faster charge transfer capability. Figure 8D illustrated the possible mechanism of ZnPc-*g*-TiO_2_-*g*-PNIPAM under visible light. Band gap values of TiO_2_ and ZnPc were 3.19 eV (Figure 5B) and 2.15 eV, respectively, based on our previous study [26]. Under visible light irradiation, photoelectrons were generated by ZnPc and then transferred to the conduction band (CB) of TiO_2_. O_2_ obtained photoelectrons and formed O_2_**·**^−^. Then, H_2_O_2_ and O_2_**·**^−^ formed OH**·** for pollutant removal. Therefore, ZnPc-sensitized TiO_2_ effectively improved the photocatalytic activity of the photocatalyst for the removal of organic pollutant under visible light.

## 3. Materials and Methods

### 3.1. Materials

*N*-isopropylacrylamide (NIPAM) was purchased from TCI (Shanghai, China) Development Co., Ltd. TiO_2_, 3-(methacryloxypropyl)-trimethoxysilane (MPS), triethylamine, potassium persulfate (KPS), benzoquinone (BQ) and isopropanol (IPA) were purchased by Shanghai Aladdin Bio-Chem Technology Co., Ltd. Rhodamine B (RhB), methylene blue (MB), and tetracycline hydrochloride (TC•HCl) were provided by Shanghai Macklin Biochemical Co., Ltd. Other reagents were purchased and used directly. 

### 3.2. Characterization

Powder X-ray diffraction (XRD) patterns were created with RigakuUltima VI diffractometer using Cu-Kα radiation. Fourier transform infrared spectra (FT-IR) were recorded on Perkin Elmer Frontier spectrometer. Scanning electron microscopy (SEM) was analyzed by means of a JEM-6701F and equipped with an Oxford INCA PentaFET-x3 EDS system. Thermogravimetric analysis (TG) was performed by NETZSCH STA 409PC. Photoluminescence (PL) spectra were analyzed using a PerkinElmer LS55 fluorescence spectrophotometer for solid samples. UV-vis diffuse reflectance spectroscopy was analyzed using a Shimadzu UV-3600 with MPC-3100. X-ray photoelectron spectroscopy (XPS) was performed with Thermo Scientific ESCALAB 250Xi XPS. N_2_ adsorption-desorption isotherms were operated on V-Sorb 2800TP.

### 3.3. Surface Modification of TiO_2_


The surface of TiO_2_ was modified by MPS to obtain TiO_2_-*g*-MPS. TiO_2_ (1 g, 12.5 mmol) was dispersed in 80 mL methylbenzene by ultrasonication for 20 min. When the mixture was heated to 100 °C, MPS (50 µL, 0.2 mmol) and triethylamine (60 µL, 0.4 mmol) were gradually added to the above solution, which was still stirred for 3 h. Then, the obtained TiO_2_-*g*-MPS was washed with water and dried at 55 °C for 24 h. Yield 47%.

### 3.4. PNIPAM Grafting on TiO_2_

The surface of TiO_2_ was functionalized with PNIPAM by radical polymerization to obtain TiO_2_-*g*-PNIPAM [27]. TiO_2_-*g*-MPS (500 mg) and NIPAM (250 mg, 2.2 mmol) were added to 50 mL deionized water and stirred for 30 min under nitrogen atmosphere. When heated to 70 °C, KPS solution (1 mL, 8 g/L) was added to the above solution. After stirring for 3 h, the solution was cooled down to room temperature and stirred for another 4 h. The obtained TiO_2_-*g*-PNIPAM was centrifuged and washed with deionized water. The obtained product was dried in 40 °C for 24 h. Yield 54%.

### 3.5. Immobilizing ZnPc on TiO_2_-g-PNIPAM

The detailed synthetic procedure of 2,9(10),16(17),23(24)-tetrakis[(3-carboxyacrylamide)phthalocyaninato]zinc (ZnPc) is shown in the Appendix A. TiO_2_-*g*-PNIPAM (100 mg) was dispersed into 30 mL water by ultrasound for 20 min. ZnPc (1 mg, 1 mol) was dissolved in 1 mL dimethyl sulfoxide and then added dropwise to the above solution. After stirring for 24 h, the obtained ZnPc-*g*-TiO_2_-*g*-PNIPAM was washed with water and ethanol. The yield was 41%.

### 3.6. Photocatalytic Experiments 

The photocatalytic activities of photocatalysts were investigated by the removal of RhB, MB and TC•HCl under visible light. The visible light source was generated by a 300 W Xe lamp equipped with a 420 nm cut-off filter. The amount of photocatalysts was 0.3 g/L. The initial concentrations of RhB, MB and TC•HCl were 4.8 mg/L, 8.0 mg/L and 10 mg/L, respectively. Before irradiation, suspensions were stirred for 30 min to reach the equilibrium of adsorption-desorption in dark and H_2_O_2_ (1 mL) used as control. After illumination, a sample (3 mL) was taken from the suspension and the concentration was monitored by UV-2600 spectrophotometer. The characteristic absorption wavelength of RhB, MB and TC•HCl were at 554 nm, 664 nm and 357 nm [28,29].

## 4. Conclusions

A thermo-responsive ZnPc-*g*-TiO_2_-*g*-PNIPAM photocatalyst sensitized with phthalocyanines was prepared. Above LCST (26 °C), it would accelerate to separate photocatalyst from solution and had effect on the photocatalytic activity. ZnPc-grafted on TiO_2_ would broaden the light response range to the visible light region and improve the separation of electron–hole pairs. ZnPc-*g*-TiO_2_-*g*-PNIPAM exhibited high photocatalytic activity for RhB and MB removal, about 97.2% and 88.6% at 20 °C in 40 min, respectively. Meanwhile, temperature has an obvious effect on the removal of RhB, and the removal rate of RhB was reduced from 97.2% to 43.4% with temperature increasing from 20 °C to 45 °C. Furthermore, ZnPc-*g*-TiO_2_-*g*-PNIPAM also shows good recycle stability. Therefore, it is believed that the photocatalyst constructed by PNIPAM, TiO_2_ and ZnPc would have an important application in water treatment.

## Data Availability

Not applicable.

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
