# Peer review of "Thermo-Responsive ZnPc-g-TiO2-g-PNIPAM Photocatalysts Sensitized with Phthalocyanines for Water Purification under Visible Light"

_molecules, 2022, doi:10.3390/molecules27103330_

Round 1

Reviewer 1 Report

The artcile concernes interisting topic about TiO2 sensibilization. There are a few points which can be improved:

  1. I think it's worth adding a description of XPS peaks of O and N elements
  2. Please add a scale in Fig. S1. Also, there are not EDS spectrum, which are mentioned in the caption. 
  3. How were the specific surface areas of 64.38 m2/g and 35.55 m2/g defined? When using the BET method, this is too high accuracy. 
  4. Please add label to axis Y in Fig. 5a. Acording 5b: the tangent line for modified TiO2 (blue line) is incorrectly. Actually, I do not think one can define the band gap in this case.
  5. Line 180-181: "the removal efficiency could reach 31.9 % in 75 min under visible light in Fig. S6". But in Fig. S6 one can see the increase in adsorption after 75 min. Please clarify.
  6. I would recommend to add information about errors to the figures presented catalytic results.
  7. Experimental details about reactor for photocatalytic tests are necessary. 

Round 2

Reviewer 1 Report

The article can be published

Reviewer 2 Report

The authors responded to the comments  and revised the manuscript and it is ready to be published.